# Effect of Secondary Paper Sludge on Physiological Traits of *Lactuca sativa* L. under Heavy-Metal Stress

**DOI:** 10.3390/plants13081098

**Published:** 2024-04-14

**Authors:** Marija Yurkevich, Arkadiy Kurbatov, Elena Ikkonen

**Affiliations:** Institute of Biology of the Karelian Research Centre, Russian Academy of Sciences, 185910 Petrozavodsk, Russia; svirinka@mail.ru (M.Y.); arkadiy1416@gmail.com (A.K.)

**Keywords:** Pb, pulp and paper sludges, soil properties, lettuce, photosynthesis, respiration

## Abstract

To eliminate the negative effect of soil contamination with heavy metals on plant growth and crop yield, different methods and techniques are the subject of discussion and study. In this study, we aimed to evaluate the effect of secondary pulp and paper-mill sludge application to soil on the response of the main physiological processes such as the growth, photosynthesis, and respiration of lettuce (*Lactuca sativa* L.) plants to soil contamination with Pb. For the pot experiment, Pb was added to sandy loam soil at concentrations of 0, 50, and 250 mg Pb(NO_3_)_2_ per kg of the soil, and secondary sludge was added to a 0, 20, or 40% sludge solution during each plant watering. The Pb-mediated change in plant biomass allocation, decrease in the photosynthetic rate, increase in leaf respiration rate, and the degree of light inhibition of respiration were closely associated with increases in both root and shoot Pb content. For the Pb-free soil condition, secondary sludge application contributed to the allocation of plant biomass towards a greater accumulation in the shoots than in the roots. Although stomatal opening was not affected by either Pb or sludge, sludge application increased photosynthetic CO_2_ assimilation regardless of soil Pb content, which was associated with an increase in the electron-transport rate and carboxylase activity of Rubisco. Soil contamination with Pb significantly increased the ratio of respiration to photosynthesis, reflecting a shift in the carbon balance toward carbon losses in the leaves, but sludge application modified the coupling between the processes with a decrease in the proportion of respiratory carbon losses. The sludge-mediated recovery of the physiological processes of *L. sativa* reflected an increase in plant tolerance to soil contamination with heavy metals, the formation of which is associated with plant and soil adjustments initiated by secondary sludge application.

## 1. Introduction

Pulp and paper-mill sludges are generated from the wastewater treatments of the pulp and paper industry. For different plant species, soil types and climatic zones, the positive effect of pulp and paper sludge on soil properties and crop yield has been established [1,2]. Recent studies have shown that pulp and paper sludges cause an increase in the soil organic matter and nutrient content, soil microbial complex and its activity, and improve soil physical properties [3,4]. Plant growth and productivity are largely dependent on nutrients and water ability in soils, as well as on soil aeration, so improving the soil with pulp and paper sludges can have a positive effect on plants. In line with this assumption, the agricultural use of pulp and paper sludges has been found to have a positive effect on crop growth and yield [5,6]. However, the effect largely depends on the pulp and paper sludges’ application rate and sludge type [5,6,7]. 

Pulp and paper-mill sludges are mainly classified as primary and secondary sludge with clear distinctions between the types in terms of chemical and physical properties [2]. The biological treatment of primary sludge with microorganisms in order to reduce the charge in dissolved organic substrates results in the formation of secondary sludge [8]. Secondary sludge has a higher organic content and a higher nitrogen and phosphorus content due to the addition of nutrients used to stimulate microbial activity during secondary treatment [9]. The beneficial uses and effects of pulp and paper-sludge application in agriculture on plant productivity have been studied mainly for primary and mixed primary and secondary sludges [2,7], but much less information is available about the use of secondary sludge in this regard. Moreover, the effect of paper sludges, including secondary, has been studied mainly on crop growth and yield [5,10], which are one of the main indicators in agricultural processes, without focusing on the plant physiological traits including fundamental processes, such as photosynthesis, respiration, and their coupling.

Unlike most domestic and industrial wastewater, only a low level of pollutants, including heavy metals, was found in the pulp and paper sludges [11]. Moreover, Lister and Line [12] showed that paper sludges can be effective biosorbents for Pb and Zn ions. Heavy metals can adsorb to certain natural substances and become immobile as a result, and soil organic matter is considered an important component controlling heavy-metal adsorption in soil [13,14]. Pulp and paper-mill sludges are rich in organic matter [9], with the organic-matter content ranges from 40% for primary sludge to 50% for secondary sludge [2]. So, it can be expected that paper sludges’ application can increase the adsorption properties of soil, which in turn can positively affect plant growth under soil contamination with heavy metals. However, there has been no published evidence supporting this state.

High levels of heavy-metal accumulation in agricultural soils negatively affect plant development and crop yields contributing to threats to global food security [15]. It is also important that soil contamination with heavy metals is associated with potential risks to the environment and human health [16]. Lead (Pb), one of the most common soil pollutants, cannot be naturally removed from soil [17] and can therefore reach high levels in soils. In addition to natural sources such as igneous and sedimentary rocks, the long-term use of pesticides and fertilizers is believed to be the main source of Pb contamination in agricultural soils [18]. The irrigation of agricultural land with domestic and industrial wastewater also contributes to Pb accumulation in soils [19]. Increased soil Pb content leads to the transformation of the many physiological processes of plants with a decrease in seed germination rate, biomass accumulation and allocation, a disruption of gene expression, enzymatic activity, nutrient uptake and water balance, and an increase in the production of reactive oxygen species [15,20,21,22]. The Pb-mediated degradation of chlorophyll, as well as the decrease in the energy transformation efficiency of photosystem II, reduced electron transport, and the low activity of Calvin cycle enzymes, can be responsible for a decline in the photosynthetic CO_2_ assimilation rate under the condition of soil contamination with Pb [23,24]. Plants may also respond to soil Pb by increasing the respiration rate [25] due to an increased demand of intermediates and energy under stress conditions [26]. The stress-mediated suppression of photosynthesis, accompanied by increased respiration, reflects a shift in the balance of plant carbon towards carbon losses, negatively affecting plant growth.

As was mentioned above, improving the chemical and physical properties of soils using pulp and paper sludges can stimulate plant growth and productivity. In addition, paper-sludge application to soil may reduce heavy metals’ load on plants. To test this hypothesis, our study aimed to evaluate secondary-sludge effects on plant physiological responses using *Lactuca sativa* L. plants as the study object. This crop species is important to the human diet and is widely cultivated all over the world. As photosynthesis is the main driver for crop production, and a balance between photosynthesis and respiration is responsible for plant biomass accumulation, our study focused on these processes and associated traits of lettuce plants. We also assessed the effect of secondary sludge on soil chemical properties, which are closely related to soil fertility and plant productivity.

## 2. Results

### 2.1. Soil Properties

Increased soil Pb concentration was accompanied by a decrease in soil pH and an increase in the content of total N in the soil (Table 1). While secondary-sludge application significantly increased carbon content only at the highest soil Pb level used in this study, for P content, this tendency was recorded at all Pb rates examined. In accordance with the increase in the sludge-application rate, soil Pb concentrations decreased especially in highly contaminated soil. The sludge application decreased soil pH, but this effect was statistically significant only for Pb-free soil. No significant effects of both Pb and sludge on the content of available Ca and Mg were found in this study (*p* > 0.05).

### 2.2. Plant-Growth Parameters

According to the two-way ANOVA, the effect of soil contamination with Pb was statistically significant for the leaf number per plant and the root–shoot ratio, but not for biomass accumulation in roots or shoots. Secondary sludges, on the contrary, affected plant biomass and did not affect the number of leaves and biomass allocation (*p* > 0.05). However, sludges only increased the leaf number of the plants grown on Pb-free soil, with no effect on plants grown on Pb-rich soil (Figure 1). Sludge application tended to increase biomass accumulation in both shoots and roots regardless of the Pb content in the soil. The contamination of the soil with the heavy metal resulted in a reduced root–shoot ratio, indicating a more strong inhibitory effect of soil Pb on roots than shoots. Sludges decreased the value of the root–shoot ratio in the plants grown on the Pb-free soil, but no significant differences in these values were found between sludge-treated and untreated plants grown on the Pb-rich soil. 

### 2.3. Plant Physiological Parameters

#### 2.3.1. Photosynthesis

Both soil Pb and secondary sludges significantly affected the gross photosynthetic CO_2_ assimilation (*A*_g_) rate of lettuce plants under study. In accordance with the increase in soil Pb content, the *A*_g_ rate and α values decreased, LCP increased, and *C*_i_, *J*, *ν*_c_, *ν*_o_, *F*_0_, *F*_v_:*F*_m_, and SPAD values were not affected by Pb regardless of its content in the soil (Table 2). Secondary-sludge application led to an increase in the *A*_g_, *J*, *ν*_c_, and α, but the increases were statistically significant mainly for plants grown in the soil with 250 mg Pb(NO_3_)_2_ kg^−1^. The highest sludge-modulated increase in the net CO_2_ assimilation rate was observed for the plants grown under the condition of the highest Pb content in the soil (Figure 2). Among the studied Pb content, no statistical differences in *C*_i_:*C*_a_, *ν*_o_, *F*_o_, *F*_v_:*F*_m_, or SPAD index were found between the sludge treatments. 

#### 2.3.2. Leaf Respiration

According to the two-way ANOVA test, leaf respiration was significantly affected by Pb but not sludge application, regardless of whether respiration occurred in light or darkness. The *R*_d_ and *R*_l_ rates were significantly higher in plants grown on Pb-containing soil than on Pb-free soil (Table 2). For all treatments, the degree the of light inhibition of leaf respiration (1 − *R*_l_:*R*_d_) varied between 5 and 24%, with an average value of 14%. 

#### 2.3.3. Respiration-to-Photosynthesis Ratio

The two-way ANOVA revealed a significant effect of both Pb content and sludge dose on the ratio of respiration to photosynthesis regardless of whether respiration occurred in light or darkness (*p* > 0.05). The *R*_d_:*A*_g_ and *R*_l_:*A*_g_ ratios increased dramatically in accordance with the increase in Pb content in the soil (Figure 3), reflecting the shift of the plant carbon balance toward carbon losses. For the plants grown on Pb-free soil, and for both *R*_d_ and *R*_l_, no significant differences in the ratio were found between the sludge treatments. However, sludge application resulted in a decrease in the ratios of plants grown on the soil with both Pb levels tested, with a recovery of carbon balances to the levels close to plants grown on the Pb-free soil.

#### 2.3.4. Leaf H_2_O Exchange

No statistically significant effect of either Pb or sludge on Tr and *g*_s_, or for RWC and PWUE values, was found, regardless of soil Pb content or sludge rate. Although the Tr and *g*_s_ values tended to decrease with an increase in the soil Pb level, and tended to increase with increasing sludge dose, these changes were not supported by statistical tests (Table 2).

### 2.4. Pb Content in Shoots and Roots

Soil contamination significantly affected the concentration of Pb in both the roots and shoots of lettuce plants. As expected, the Pb content in plant organs increased in accordance with the increase in soil Pb among all sludge treatments (Figure 4). No significant differences in both Pb contents were found between the sludge treatments under the conditions of Pb-free soil. However, secondary-sludge application led to a decrease in Pb content in the roots under the highest soil Pb level used in this study, and in the shoots under the rate of 50 mg Pb(NO_3_)_2_ per kg of the soil. 

### 2.5. Leaf Mass per Area and Area-Based Leaf Nutrient Content

The ANOVA test revealed no significant effect of soil Pb on the LMA, LNC, or LPC values, but the effect of sludge application was statistically significant for LMA and LNC. Under Pb-free soil and the soil containing 250 mg Pb(NO_3_)_2_ kg^−1^, LMA increased with increasing sludge dosage (Figure 5). While both Pb and sludge had no effect on LPC, sludge application increased LNP in the soil without Pb and in the soil with 250 mg Pb(NO_3_)_2_ kg^−1^. For A_n_, a positive significant correlation was found with LNC, but not with LPC (Figure 6).

### 2.6. Photosynthetic Nutrient-Use Efficiency

Among all treatments, the highest PNUE and PPUE values were found in plants grown on Pb-free soil without sludge applications (Figure 7). Soil contamination with Pb caused a statistically significant decrease in PNUE value. For plants grown on Pb-rich soil, secondary-sludge application had no effect on either PNUE or PPUE. 

## 3. Discussion

Soil contamination with Pb can affect the availability of soil macronutrients to plants, as found in this study (Table 1). High concentrations of soil Pb were accompanied by an increase in the content of total N and available K in the soil, as well as a decrease in the level of available P. Phosphorus is involved in primary metabolic processes such as photosynthesis and respiration and plays a key role in complex energy transformations as the main structural component of ATP (adenosine triphosphate) [27]. A decreased availability of soil P may be one of the reasons for the inhibition of photosynthesis in the studied plants when exposed to high concentrations of Pb in the soil. The negative effect of Pb on the photosynthetic apparatus is well studied [28,29,30], and our study confirms the conclusions. Pb accumulation in the lettuce organs, along with a decrease in the photosynthetic rate, caused an increase in respiratory costs (Figure 2 and Figure 4, Table 3), which is consistent with early studies [24] and appears to be associated with an increase in energy and intermediate demand under the stress. Both Pb-mediated inhibition in the CO_2_ assimilation rate and increased respiration rate caused a shift in the plant carbon balance towards carbon losses (Figure 3), resulting in a change in plant biomass allocation between roots and shoots (Figure 1). 

Secondary paper-sludge application affected macronutrients content in the soil with a more pronounced impact on C and P concentration (Table 1 and Table 2). The sludge-mediated increase in soil P may be due to the introduction of phosphorus into the biological treatment processes of paper sludges, which results in an increase in the phosphorus content of the secondary sludge [2]. A positive effect of secondary pulp and paper sludge on the content of some macronutrients, including phosphorus, was also shown by Nunes et al. [5] for Mediterranean agricultural soils. Increased soil P may be partly responsible for the recovering of photosynthetic activity and stabilization of the respiration-to-photosynthesis ratio of lettuce plants grown in the Pb-rich soil. Multiple regulatory mechanisms determine plant photosynthesis and its tolerance to stress. For the lettuce plants exposed to high Pb concentrations in the soil, our study identified electron-transport and carboxylation rates as modifying traits that improve the rate of CO_2_ assimilation under the secondary-sludge application. The sludge-mediated increase in *A*_n_ was accompanied by an increase in LMA, indicating structural changes in leaf tissue that may be responsible for enhanced photosynthesis. The *A*_n_ rate was also positively associated with area-based leaf N content (Figure 6), supporting earlier studies [31,32]. The positive correlation occurs because most of the N is integrated into photosynthetic proteins, especially Rubisco [30]. The net photosynthetic rate per unit N in a leaf, termed photosynthetic nitrogen-use efficiency, has been considered an important functional trait in characterizing leaf physiology [33]. The PNUE values were drastically declined by both soil Pb and sludge application (Figure 7). Numerous studies have shown a negative correlation between PNUE and LMA [30,33], and our results are in agreement with this finding (Figure 5 and Figure 7). The higher LMA in plants grown under sludge application can relate to a higher fraction of leaf dry mass and cell walls. Greater N partitioning into cell-wall proteins against lesser N partitioning into Rubisco proteins can be responsible for the negative relation between LMA and PNUE [32]. Given the negative relationship between *A*_n_ and PNUE and the positive relationship between *A*_n_ and LMA, the sludge-mediated increase in the net CO_2_ assimilation rate may be due to structural changes in leaf tissue as well as carboxylation capacity, but not to an increase in photosynthetic N-use efficiency.

The coupling between the photosynthetic and respiration processes is an important indicator of the carbon balance of plants and is useful in assessing the ability of plants to adapt to stress conditions [34]. The stabilized ratio of respiration to photosynthesis under the sludge application, whether respiration occurs in light or darkness, may be responsible for improving lettuce growth on the Pb-contaminated soil. The leaf respiration in the light is considered a more stable process than respiration in darkness [35]. Our results revealed high variability in both *R*_l_ and *R*_d_ rates under zero and low soil Pb content, but under a high Pb level, the effect of sludge-amendment rate was greater for *R*_d_ than *R*_l_. This caused a change in the 1 − *R*_l_:*R*_d_ value, which reflects the extent of the light inhibition of respiration, with a decrease in inhibition following an increase in the sludge-amendment rate.

Not only the chemical but also the physical properties of soils can be transformed when introducing secondary paper sludge into soils [4]. Soil texture and structure play an important role in soil water availability for plants and nutrient uptake [36] and therefore successful plant growth and productivity. Secondary-sludge application when watering plants can lead to the formation of stable organo-mineral particles with an increase in the proportion of large drainage pores and saturated water content, providing increased water supply to plants [4,37]. The secondary sludge decreased the effective thermal conductivity of the soil in an air-dried state but increased the heat transfer in wet soil, which may be associated with the sludge-mediated increase in the soil water content [4]. Improving the physical properties of the soil using secondary sludge, as well as the chemical properties, could have a positive effect on the physiological traits of lettuce plants under the study.

The lettuce plants grown in the Pb-rich soil displayed elevated Pb contents in both the roots and leaves (Figure 4) with, as expected, a higher metal level in leaves than in roots. Secondary-sludge application provided a reduction in Pb levels in roots under the highest soil Pb content for this study. Experimental evidence supports the role of soil organic matter in reducing the mobility and phytoavailability of heavy metals including Pb, by the formation of organo-mineral complexes [38,39]. Moreover, Pb mobility can be reduced by phosphate application into soil [39,40]. Since the secondary sludge contains organic matter and phosphorus, its application to the soil could contribute to Pb immobilization reducing its uptake by plants, thereby helping to reduce heavy-metal stress on the plants.

## 4. Materials and Methods

### 4.1. Plant Materials Preparation

The sandy loam soil used for this pot experiment was collected randomly from the topsoil layer in the northwest region of Russia (61.826573, 33.179712). The collected soil is characterized by a low natural fertility, low water holding capacity, and low pH (5.46) and humus content (0.5–2.5%), and had 0.39% of total N and 1.6 g kg^−1^ of available P and K [41]. The available Pb content of collected soil was 0.13 ± 0.01 mg kg^−1^. After air drying, the soil was sieved with a 2 mm sieve, then fertilized with 150 mg kg^−1^ of N, P, and K before being mixed with Pb(NO_3_)_2_ at a rate of 0, 50, or 250 mg kg^−1^. The soil substrates were well watered and incubated under 70–80% of the maximum soil water-holding capacity and air temperature of 21 ± 1 °C for two weeks, and then placed in 0.8 L pots (D = 15 cm) with a soil bulk density of about 1.4 g cm^−3^. Each Pb treatment included 24 pots that were divided into three parts and served as units for secondary-sludge treatments. 

Six seeds of lettuce (*Lactuca sativa* L. var. Medvezhje ushko) were sown per pot. All pots were subjected to controlled conditions of 23/20 °C day/night temperature, a 16 h photoperiod, and 250 μmol m^−2^ s^−1^ of photosynthetic photon flux density (PPFD). The pots were randomly repositioned daily to avoid plant-location effects within the location area. The pots were watered through the soil surface every two days following a design for secondary-sludge treatment. The first part of the pots of each Pb treatment was watered with distilled water (0% treatment), and the second and third parts were watered with 20% and 40% solutions of the secondary pulp and paper-mill sludges (20% and 40% treatment, accordingly). The sludge concentrations were chosen in a preliminary experiment. In this way, the completely randomized experimental design included three levels of Pb content in the soil (0, 50, or 250 mg Pb(NO_3_)_2_ kg^−1^) and three sludge treatments for each Pb treatment. Each unit treatment included eight pots. The design and execution of the experiment followed the strategies recently reviewed [42]. During the 46-day experiment, 0, 139, and 278 mL of the secondary sludge was added to 1 L of soil, accordingly, for 0%, 20%, and 40% treatments. The sludge used in this study was generated after the secondary aeration of wastewater from a pulp-and-paper mill located in the northwestern region of Russia. The chemical and physical characteristics of secondary sludge used in the study are presented in Table 3. To achieve a sludge content of 20 and 40%, it was mixed with distilled water. Two weeks after sowing, the seedlings were thinned to three per pot. 

### 4.2. Plant-Growth Parameters

For each treatment, four 17-, 21-, 34-, and 46-day-old lettuce plants were harvested and separated into shoots and roots. The roots were carefully separated from the soil and washed. The total leaf number per plant was counted. The collected shoots and roots were dried under a temperature of 70 °C to a constant weight and weighed. The ratio of root dry biomass to shoot dry biomass (root–shoot) was calculated. The total leaf area per plant was determined by leaf scanning and using the program “AreaS”. The leaf mass per area (LMA) was calculated as dry leaf mass divided by leaf area.

### 4.3. Gas-Exchange Parameters

The leaf area-based CO_2_ and H_2_O gas-exchange parameters were measured on mature *L. sativa* leaves of 43–46-day plants using a portable photosynthesis system (HCM-1000, Walz, Effeltrich, Germany). The leaves were randomly selected from plants from different pots of each treatment. During each measurement, the temperature and air humidity in the leaf chamber were maintained at 20 °C and 60–70%, respectively. The net CO_2_ assimilation rate (*A*_n_), stomatal conductance (*g*_s_), transpiration water losses (Tr), and intercellular (*C*_i_) and ambient (*C*_a_) CO_2_ concentration were determined at 1200 μmol m^−2^ s^−1^ of PPFD. In addition, to determine the light-response curves of photosynthesis, the *A*_n_ rate was measured at 300, 2000, 1200, 1000, 800, 300, 60, 40, 20, and 0 μmol m^−2^ s^−1^ of PPFD. Readings were taken after a steady-state rate of *A*_n_ was reached. The apparent quantum yield of photosynthesis (α) was calculated as the slope of *A*_n_ versus irradiances of 20, 40 and 60 μmol PPFD m^−2^ s^−1^, according to Garmash and Golovko [43]. The photosynthetic electron-transport rate (*J*), carboxylase activity of Rubisco (*v*_c_), and oxygenase activity of Rubisco (*v*_o_) were calculated according to Farquhar and von Caemmerer [44]. The leaf respiration rate in darkness (*R*_d_) was taken after 30 min of zero irradiance. The rate of leaf respiration in the light (*R*_l_) was calculated according the Kok method [45]. The gross photosynthesis rate (*A*_g_ = *A*_n_ + *R*_l_) with *A*_n_ at 1200 μmol m^−2^ s^−1^ of PPFD, and the ratios *R*_d_ to *A*_g_ (*R*_d_:*A*_g_), *R*_l_ to *A*_g_ (*R*_l_:*A*_g_), and *R*_l_ to *R*_d_ (*R*_l_:*R*_d_), were calculated. The light-compensation point (LCP), which is the light level at which the rate of photosynthesis is equal to the rate of respiration, was determined using the light-response curve of photosynthesis. The photosynthetic water-use efficiency (PWUE) value at the leaf level was defined as the *A*_n_-to-Tr ratio. 

### 4.4. Chlorophyll Fluorescence and Content; Relative Water Content of Leaves

Chlorophyll-fluorescence parameters and chlorophyll content were measured on the same leaves as the gas-exchange parameters. The minimum and maximum fluorescence (*F*_o_ and *F*_m_, respectively) and the maximum photochemical quantum efficiency of PSII (*F*_v_:*F*_m_) of leaves were measured using MINI-PAM (Walz, Effeltrich, Germany). The leaves were previously dark-adapted for 30 min. The *F*_v_:*F*_m_ values were calculated as (*F*_m_ − *F*_o_):*F*_m_. The chlorophyll content was measured using a SPAD-502 chlorophyll meter (Minolta, Tokyo, Japan). 

For measuring relative water content (RWC), leaf discs were collected and the fresh weight (FW) of the disks was determined. After that, the disks were placed in distilled water at 4 °C for 24 h for rehydration. The leaf turgid weight (TW) was determined after the incubation period, and leaf disks were dried at 70 °C before the constant weight. The dry weight (DW) of leaf disks was measured, and the RWC value was calculated as (FW − DW)/(TW − DW) × 100%, according to González and González-Vilar [46].

### 4.5. Chemical Analyses

Samples of soil and plant organs (shoots and roots) from each treatment were collected after the completion of the measurements of plant physiological parameters. The soils from five pots of each treatment were collected and mixed, and then three subsamples were used for chemical analyses. All plants of each treatment were collected from all pots, and plant organs were mixed and divided into three repetitions. The analyses were carried out in the Core Facility “Analytical laboratory” of the Forest institute of KRC of RAS. For C and N analysis, the soil samples were sieved with a less-than-0.05 mm sieve. The pH of each soil sample was measured in 1 M L^−1^ KCl solution (pHKCl) at a ratio of 1:2.5 (w:w). Total soil carbon was analyzed with a total organic C analyzer (TOC-L CPN, Shimadzu, Japan). The leaf samples of 0.2–0.3 g were preliminarily homogenized and digested with HNO_3_ and HCl (Vekton, Saint Petersburg, Russia). The total N content in the samples was determined by the Kjeldahl method [47]. The available P in soil samples was extracted with 0.2 M HCl. Soil Ca and Mg were extracted with 1MNH_4_Ac buffered to pH 7. The mass-based concentrations of Pb in the plant tissue and soil were determined by spectrophotometric atomic absorption (Shimadzu AA-7000, Kyoto, Japan). The N content in plant samples was determined by the nesslerization of the ammonia, and the P content was determined by the ammonium molybdate method. Spectrophotometric and flame photometric methods were used to measure mass-based N and P, accordingly (SF 2000 OKB Spectrum, Saint Petersburg, Russia). Area-based leaf N concentration (LNC) and P concentration (LPC) were calculated using the values of LMA and the mass-based element contents. Photosynthetic nitrogen-use efficiency (PNUE) and photosynthetic phosphorus-use efficiency (PPUE) were calculated by dividing *A*_n_ by LNC and LPC, accordingly. 

### 4.6. Statistical Analysis

For each treatment, the means ± SE were determined with at least five replicates for the physiological parameters of plants and three replicates for chemical analyses. To evaluate the significant differences between the means, the LSD test at a *p* < 0.05 level was used (Statistica software, v. 8.0.550.0, StatSoft, Inc., Tulsa, OK, USA). 

## 5. Conclusions

In summary, this study demonstrated the benefits of using secondary paper sludge for *L. sativa* in optimizing plant carbon balance, especially under soil contamination with Pb. The sludge application contributed to the recovery of the photosynthetic activity of plants grown in Pb-rich soil to the level of their counterparts grown in Pb-free soil with the stabilization of coupling between photosynthesis and respiration. The sludge-mediated recovery of photosynthetic capacity was mainly attributed to structural adjustments and enhanced carboxylation capacity, but not to an increase in photosynthetic nutrient-use efficiency. The positive effect of secondary sludge on plants could presumably be achieved both through improving the chemical and physical properties of the soil as well as through the immobilization of Pb in the soil. 

## Figures and Tables

**Figure 1 plants-13-01098-f001:**
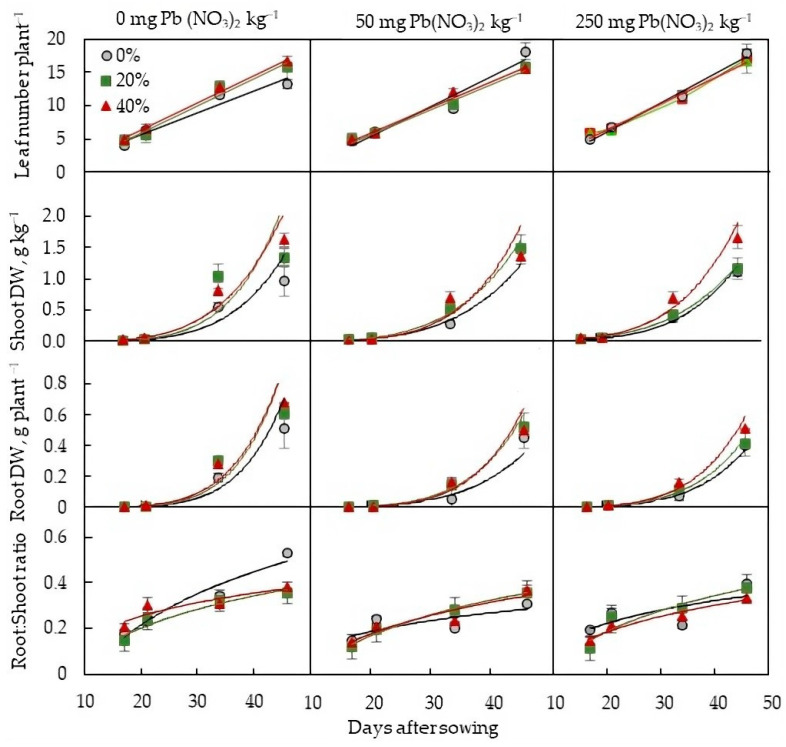
Leaf number per plant, shoot dry weight (DW), root dry weight, and root–shoot ratio of *L. sativa* grown on the soil containing 0, 50, or 250 mg Pb(NO_3_)_2_ kg^−1^ and watered with 0, 20, or 40% secondary pulp and paper-mill sludge.

**Figure 2 plants-13-01098-f002:**
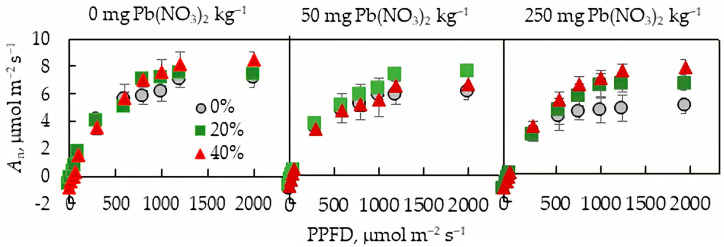
Light curve of net CO_2_ assimilation rate of *L. sativa* grown in the soil containing 0, 50, or 250 mg Pb(NO_3_)_2_ kg^−1^ and watered with 0 (grey), 20 (green), or 40 (red)% secondary pulp and paper-mill sludge.

**Figure 3 plants-13-01098-f003:**
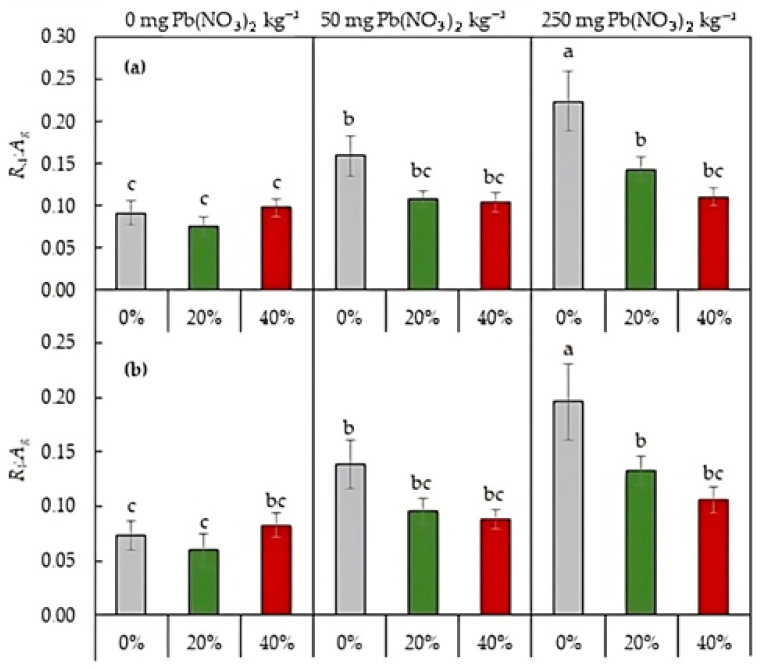
Ratios of leaf respiration in darkness (*R*_d_, **a**) or in the light (*R*_l_, **b**) to gross CO_2_ assimilation rate (*A*_g_) of *L. sativa* grown in the soil containing 0, 50, or 250 mg Pb(NO_3_)_2_ kg^−1^ and watered with 0 (grey), 20 (green), or 40 (red)% secondary pulp and paper-mill sludge. Different letters indicate significant differences between the means.

**Figure 4 plants-13-01098-f004:**
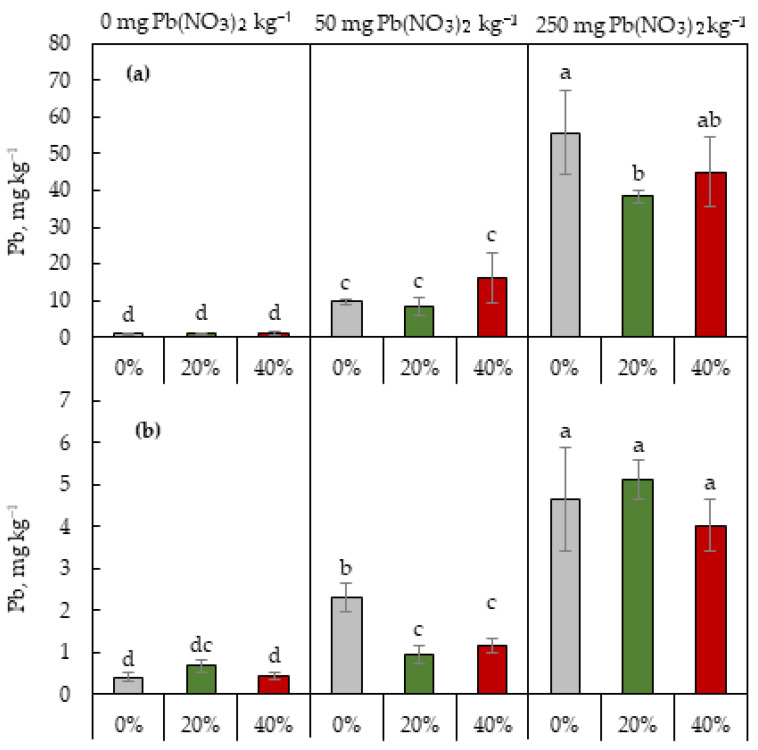
Pb content in roots (**a**) and shoots (**b**) of *L. sativa* plants grown on the soil containing 0, 50, or 250 mg Pb(NO_3_)_2_ kg^−1^ and watered with 0 (grey), 20 (green), or 40 (red)% secondary pulp and paper-mill sludge. Different letters indicate significant differences between the means.

**Figure 5 plants-13-01098-f005:**
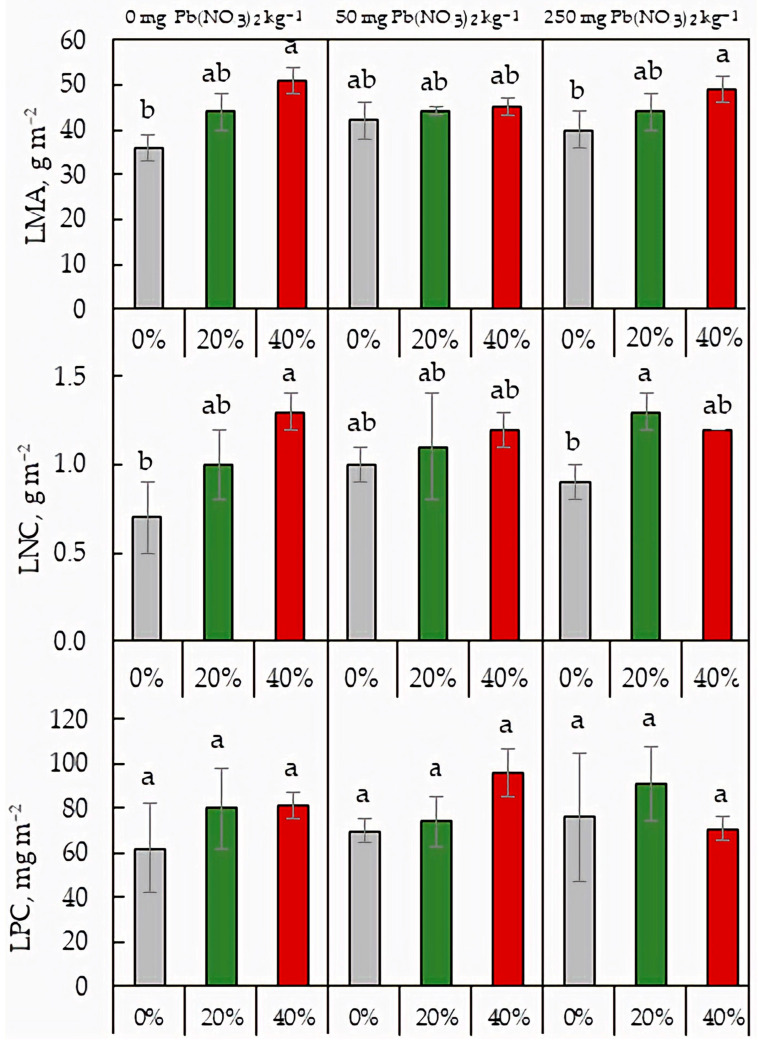
Leaf mass per area (LMA), area-based leaf N content (LNC), and area-based leaf P content (LPC) in *L. sativa* plants grown on the soil containing 0, 50, or 250 mg Pb(NO_3_)_2_ kg^−1^ and watered with 0 (grey), 20 (green), or 40 (red)% secondary pulp and paper-mill sludge. Different letters indicate significant differences between the means.

**Figure 6 plants-13-01098-f006:**
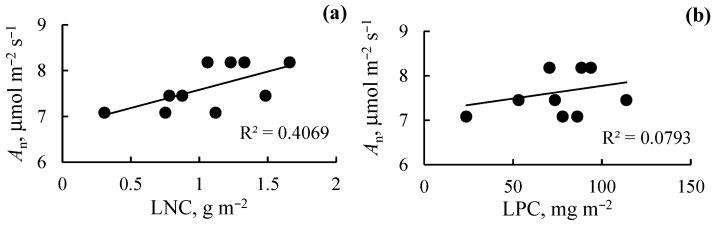
Relationship between net CO_2_ assimilation rate (*A*_n_), area-based leaf N content (LNC, **a**), and area-based leaf P content (LPC, **b**) in *L. sativa*.

**Figure 7 plants-13-01098-f007:**
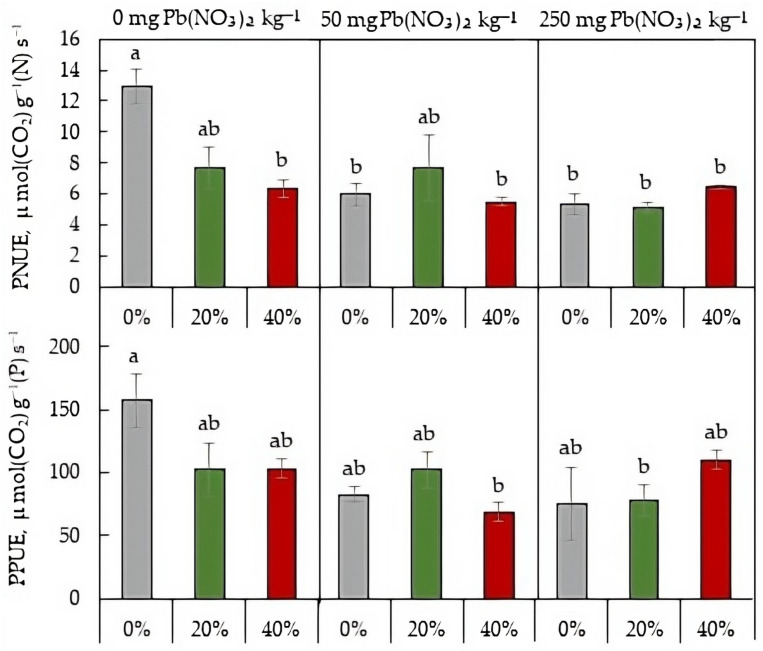
Photosynthetic nitrogen-use efficiency (PNUE) and photosynthetic phosphorus-use efficiency (PPUE) in *L. sativa* grown in soil containing 0, 50, or 250 mg Pb(NO_3_)_2_ kg^−1^ and watered with 0 (grey), 20 (green), or 40 (red)% secondary pulp and paper-mill sludge. Different letters indicate significant differences between the means.

**Table 1 plants-13-01098-t001:** Chemical and physical properties of the soil substrate containing 0, 50, or 250 mg Pb(NO_3_)_2_ kg^−1^ and watered with the 0, 20, or 40% of secondary pulp and paper-mill sludge.

Parameter	0 mg Pb(NO_3_)_2_ kg^−1^			50 mg Pb(NO_3_)_2_ kg^−1^			250 mg Pb(NO_3_)_2_ kg^−1^		
	0%	20%	40%	0%	20%	40%	0%	20%	40%
total C, %	4.3 ± 0.1 abc	4.4 ± 0.1 abc	4.7 ± 0.1 a	4.2 ± 0.2 b	4.5 ± 0.2 ab	4.4 ± 0.1 abc	4.0 ± 0.2 c	4.3 ± 0.1 abc	4.7 ± 0.2 a
total N, %	0.27 ± 0.00 c	0.29 ± 0.00 b	0.28 ± 0.00 b	0.29 ± 0.01 b	0.28 ± 0.00 b	0.28 ± 0.01 b	0.40 ± 0.00 a	0.41 ± 0.00 a	0.40 ± 0.00 a
available P, g kg^−1^	0.91 ± 0.04 c	1.16 ± 0.08 a	1.26 ± 0.02 a	0.91 ± 0.06 c	1.00 ± 0.09 bc	1.14 ± 0.06 ab	0.88 ± 0.03 c	0.85 ± 0.02 c	0.95 ± 0.04 c
available Mg, mg kg^−1^	196 ± 2 abc	196 ± 9 abc	198 ± 6 abc	212 ± 8 a	189 ± 10 c	190 ± 1 bc	193 ± 10 abc	210 ± 4 ab	202 ± 6 abc
available Ca, g kg^−1^	1.95 ± 0.05 bc	2.07 ± 0.06 bc	2.22 ± 0.06 ab	1.93 ± 0.24 bc	1.86 ± 0.10 c	1.99 ± 0.06 bc	2.15 ± 0.08 abc	2.34 ± 0.03 a	2.39 ± 0.05 a
available Pb, mg kg^−1^	0.1 ± 0.0 c	0.1 ± 0.0 c	0.1 ± 0.0 c	4.0 ± 0.5 c	2.8 ± 0.1 c	3.0 ± 0.5 c	41.6 ± 3.6 a	45.4 ± 2.3 a	36.6 ± 1.5 b
pH	5.40 ± 0.03 a	5.28 ± 0.04 b	5.26 ± 0.03 bc	5.25 ± 0.02 bc	5.19 ± 0.02 c	5.19 ± 0.01 c	5.29 ± 0.02 b	5.25 ± 0.01 bc	5.28 ± 0.03 b

Different letters indicate significant differences between the means.

**Table 2 plants-13-01098-t002:** Physiological traits of *L. sativa* grown on the soil containing 0, 50, or 250 mg Pb(NO_3_)_2_ kg^−1^ and watered with 0, 20, or 40% secondary pulp and paper-mill sludge.

Parameter	0 mg Pb(NO_3_)^2^ kg^−1^			50 mg Pb(NO_3_)^2^ kg^−1^			250 mg Pb(NO_3_)^2^ kg^−1^		
	0%	20%	40%	0%	20%	40%	0%	20%	40%
*A*_g_, μmol m^−2^ s^−1^	7.6 ± 0.5 abc	7.9 ± 0.6 ab	8.9 ± 0.6 a	6.6 ± 0.5 bc	7.6 ± 0.5 abc	7.5 ± 0.7 abc	5.9 ± 0.4 c	7.7 ± 0.5 abc	8.7 ± 0.4 a
*C*i*:C*a	0.59 ± 0.04 ab	0.55 ± 0.05 ab	0.59 ± 0.03 ab	0.61 ± 0.02 ab	0.56 ± 0.02 ab	0.53 ± 0.04 b	0.62 ± 0.07 ab	0.63 ± 0.03 a	0.60 ± 0.04 ab
*J*, µmol m^−2^ s^−1^	48 ± 4 b	53 ± 3 ab	58 ± 3 a	46 ± 4 ab	53 ± 4 ab	48 ± 5 ab	43 ± 6 b	46 ± 3 b	55 ± 4 a
*ν*_c_, µmol m^−2^ s^−1^	8.9 ± 0.6 b	9.7 ± 0.7 ab	10.8 ± 0.6 a	8.7 ± 0.7 ab	9.7 ± 0.6 ab	9.1 ± 0.9 ab	8.1 ± 1.0 b	8.7 ± 0.8 ab	10.4 ± 0.7 a
*ν*_o_, µmol m^−2^ s^−1^	1.9 ± 0.3 a	1.7 ± 0.4 a	2.2 ± 0.4 a	2.1 ± 0.1 a	1.7 ± 0.1 a	2.1 ± 0.3 a	1.9 ± 0.3 a	2.2 ± 0.2 a	2.4 ± 0.2 a
α, µmol µmol^−1^	0.018 ± 0.001 a	0.020 ± 0.002 a	0.018 ± 0.001 a	0.017 ± 0.001 ab	0.017 ± 0.001 a	0.018 ± 0.001 a	0.014 ± 0.002 b	0.017 ± 0.001 ab	0.018 ± 0.001 a
*F* _o_	449 ± 8 a	427 ± 8 a	417 ± 20 a	449 ± 35 a	413 ± 9 a	427 ± 15 a	396 ± 22 a	419 ± 14 a	437 ± 23 a
*F*_v_:*F*_m_	0.844 ± 0.004 a	0.846 ± 0.004 a	0.834 ± 0.012 a	0.827 ± 0.014 a	0.839 ± 0.007 a	0.843 ± 0.009 a	0.842 ± 0.007 a	0.833 ± 0.009 a	0.826 ± 0.016 a
SPAD index	15.3 ± 0.6 ab	16.1 ± 0.9 a	14.2 ± 0.5 ab	14.1 ± 0.6 ab	14.5 ± 0.6 ab	15.8 ± 1.6 a	14.0 ± 1.1 ab	13.0 ± 0.5 b	13.0 ± 0.8 b
LCP, µmol m^−2^ s^−1^	33 ± 5 cd	24 ± 6 d	42 ± 7 bcd	54 ± 6 b	43 ± 5 bcd	37 ± 4 bcd	78 ± 15 a	59 ± 6 b	52 ± 9 c
*R*_d_, µmol m^−2^ s^−1^	0.71 ± 0.07 bc	0.61 ± 0.13 c	0.87 ± 0.11 ab	0.99 ± 0.08 ab	0.90 ± 0.07 bc	0.74 ± 0.06 b	1.14 ± 0.18 a	1.07 ± 0.09 a	0.97 ± 0.14 ab
*R*_l_, µmol m^−2^ s^−1^	0.58 ± 0.08 bc	0.49 ± 0.15 c	0.74 ± 0.11 abc	0.86 ± 0.08 ab	0.73 ± 0.09 abc	0.64 ± 0.07 bc	1.00 ± 0.17 a	0.99 ± 0.09 a	0.93 ± 0.15 ab
*R*_l_:*R*_d_	0.81 ± 0.04 b	0.76 ± 0.09 b	0.83 ± 0.04 ab	0.87 ± 0.01 ab	0.87 ± 0.03 ab	0.86 ± 0.04 ab	0.87 ± 0.01 ab	0.93 ± 0.03 a	0.95 ± 0.05 a
1 − *R*_l_:*R*_d_, %	19 ± 4 a	24 ± 9 a	17 ± 4 ab	13 ± 1 ab	13 ± 1 ab	16 ± 3 ab	13 ± 1 ab	7 ± 3 a	5 ± 2 a
Tr, mmol m^−2^ s^−1^	1.1 ± 0.1 a	1.0 ± 0.2 a	1.2 ± 0.1 a	0.9 ± 0.1 a	1.1 ± 0.1 a	0.9 ± 0.1 a	0.9 ± 0.1 a	1.1 ± 0.1 a	1.2 ± 0.1 a
*q*_s_, mmol m^−2^ s^−1^	81 ± 9 a	73 ± 11 a	94 ± 9 a	69 ± 7 a	78 ± 8 a	69 ± 12 a	70 ± 10 a	82 ± 6 a	89 ± 7 a
RWC, %	71 ± 2 a	69 ± 3 a	70 ± 1 a	69 ± 2 a	73 ± 2 a	72 ± 1 a	76 ± 1 a	70 ± 2 a	70 ± 2 a
PWUE, µmol mmol^−1^	6.8 ± 0.7 a	7.3 ± 0.7 a	6.8 ± 0.2 a	6.4 ± 0.6 a	6.6 ± 0.3 a	7.2 ± 0.7 a	6.4 ± 0.9 a	6.4 ± 0.7 a	6.6 ± 0.7 a

Different letters indicate significant differences between the means.

**Table 3 plants-13-01098-t003:** Chemical and physical properties of secondary sludge under the study.

C, %	pH	N, mg L^−1^	P, mg L^−1^	K, mg L^−1^	Na, mg L^−1^	Ca, mg L^−1^	Mg, mg L^−1^
56.7	7.16	238	64	9	47	233	22

## Data Availability

The original contributions presented in the study are included in the article, further inquiries can be directed to the corresponding author.

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
