# Peer review of "Effect of Secondary Paper Sludge on Physiological Traits of Lactuca sativa L. under Heavy-Metal Stress"

_plants, 2024, doi:10.3390/plants13081098_

Round 1

Reviewer 1 Report

Comments and Suggestions for Authors

The submitted article is very interesting and indicates an interesting possibility of improving the quality of the soil in which lettuce can be grown. The article is written in a logical and orderly manner. The only thing that is puzzling is the fact that the addition of secondary paper sludge improves the plant yield but does not reduce Pb uptake. In some cases, the concentration of lead in the plants was even higher than in plants growing on soils without added sediment. Isn't there a risk that we will get nice plants, but they will be contaminated with lead, which means they will pose a threat to the health of people who consume them? What is the point of growing such plants?

Author Response

First, we would like thank the Reviewer for attentive revision of our manuscript and principal question.

The submitted article is very interesting and indicates an interesting possibility of improving the quality of the soil in which lettuce can be grown. The article is written in a logical and orderly manner. The only thing that is puzzling is the fact that the addition of secondary paper sludge improves the plant yield but does not reduce Pb uptake. In some cases, the concentration of lead in the plants was even higher than in plants growing on soils without added sediment. Isn't there a risk that we will get nice plants, but they will be contaminated with lead, which means they will pose a threat to the health of people who consume them? What is the point of growing such plants?

Yes, you are right, today there is a risk to human health from agricultural food products. Unfortunately, at present, significant areas of agricultural soils are contaminated, including with heavy metals, and these areas continue to be used for agricultural purposes. There are many examples of such use, but I show only one. The Mezquital Valley, central Mexico, has been used to grow crops for hundreds of years. Since 1900, the fields have been irrigated with industrial and domestic wastewater from the Mexico City. Irrigation with wastewater significantly increased crop yields, but the long period of fertilization has affected soil properties with soil salinization and contamination, including with heavy metals. The population cannot stop cultivating these fields because there is no more free land. The problem of contamination of agricultural soils with heavy metals is being solved from different points of view, and we believe our study make contribution. I would like to note that the permissible level of lead in cultivated plants is 5 mg/kg fresh weight. In our experiment the lead content is lower than 5 mg/kg dry weight.

To clarify the problem, we added in the text: ‘In addition to natural sources such as igneous and sedimentary rocks, long-term use of pesticides and fertilizers is believed to be the main source of Pb contamination in agricultural soils [18]. Irrigation of agricultural land with domestic and industrial wastewater also contributes to the accumulation of Pb in soils [19].’ Lines 72-76.

All changes to the text are highlighted in yellow.

Thank you once more,

Best regards,

Reviewer 2 Report

Comments and Suggestions for Authors

The manuscript entitled "Recovery effect of secondary paper sludge on physiological traits of Lactuca sativa L. under heavy metal stress" presents the effects of applying secondary sludge through irrigation to lettuce plants exposed to toxic levels of lead, on the growth, photosynthetic machinery, lead accumulation in tissues as well as on soil chemical properties. The results concerning the photosynthetic parameters do not evidence considerable significant effects either from Pb or sludge application. Still, the authors report trends of increase/decrease that are not supported by the statistics presented in the tables and graphs. This makes the interpretation extremely confusing. It becomes more evident that the main beneficial effects of sludge application are related to the improvement of soil physical properties, which were not the aim of this work, improved root biomass production and a decrease in lead availability and uptake. The authors should present only the statistically significant results and direct the discussion toward those effects only, making the impacts of those results more clear to the readers. It would be a very nice addition to measure the elemental composition (P, N, K, Ca, Mg) in leaf and root tissues as well. The authors should also measure parameters related to carbon metabolism, given the observed trends in carbon fixation or loss. The content of soluble sugars and proteins would already provide important information. Analysis of the activity of relevant enzymes from primary metabolism would also improve this work. The manuscript should be accepted only if majorly revised, with attention to these and the following comments:

Specific comments:

Pay attention to the use of abbreviations. For example, the word lead appears several times in the text instead of Pb.

Line 13: Replace "with the rate of" with "at the concentrations of"

Line 14: Rephrase "with each watering plants". This is not understandable.

Line 47: Their use to initiate microbial activity? Or their production from the microbial activity? Clarify.

Line 49: What do the authors mean with "mixed"?

Lines 69-70: The authors could explore further on the main ways of input of Pb into agricultural soils.

Line 103: Remove "regardless of Pb content". This is not significant for plants grown under Pd-contaminated soils.

Lines 125-126: This is not true. This is only visible for Pb-free treatment.

Lines 129-130: This is not true. Replace increased with decreased.

Lines 148-149: Do not present results that are not statistically significant.

Lines 149-150: If this is one of the few truly significant results, it should be further explored and discussed.

In table 2, add a row with the values of 1-Rl:Rd.

Lines 165: Replace affected with increased.

Line 168: Do not describe trends that are not supported by statistical significance.

Line 170: "varied between 5 and 39%, with an average value of 17%." Why is this interesting to present?

Line 171: "highest values of Rl:Rd" This values were not significantly higher than in plants grown on Pb-free soil.

Line 172-173: Not statistically significant.

Line 281: Please provide the diameter or area of each pot.

Line 288: How were pots irrigated? From the bottom or at the surface?

Table 4 does not have any title.

Line 304: Harvested

Line 359-361: Most elements were not quantified in the plant tissues, right?

Line 366: Two-way ANOVA, not One-way.

Line 368-369: This conclusion seems a bit exaggerated considering the results presented. 

References should be formatted so that all species names appear italicized.

Author Response

First, we would like thank the Reviewer for attentive revision of our manuscript and constructive comments and suggestions.

Let me indicate the modifications made in the manuscript in the light of Reviewer’s comments.

Still, the authors report trends of increase/decrease that are not supported by the statistics presented in the tables and graphs. This makes the interpretation extremely confusing. It becomes more evident that the main beneficial effects of sludge application are related to the improvement of soil physical properties, which were not the aim of this work, improved root biomass production and a decrease in lead availability and uptake. The authors should present only the statistically significant results and direct the discussion toward those effects only, making the impacts of those results more clear to the readers.

Thank you. According to the commentary, the Discussion section currently only discusses statistically significant differences, with no mention of trends.

 It would be a very nice addition to measure the elemental composition (P, N, K, Ca, Mg) in leaf and root tissues as well.

Yes, we added data on macronutrients content in plant leaves and roots to the manuscript: Figure 5, 6. Also we added data on photosynthetic nutrient use efficiency: Figure 7. The section Discussion was modified according these manipulations.

The authors should also measure parameters related to carbon metabolism, given the observed trends in carbon fixation or loss. The content of soluble sugars and proteins would already provide important information. Analysis of the activity of relevant enzymes from primary metabolism would also improve this work. The manuscript should be accepted only if majorly revised, with attention to these and the following comments:

Thank you very much. Our immediate plans include measuring the parameters you reported.

Specific comments:

Pay attention to the use of abbreviations. For example, the word lead appears several times in the text instead of Pb.

We have replaced the word "lead" with "Pb" throughout the text.

Line 13: Replace "with the rate of" with "at the concentrations of"

Thank you. Done.

Line 14: Rephrase "with each watering plants". This is not understandable.

We used ‘during each plant watering’ instead of ‘with each watering plants’.

Line 47: Their use to initiate microbial activity? Or their production from the microbial activity? Clarify.

Yes. We rewrite ‘due their use to initiate microbial activity’ to ‘due to the addition of nutrients used to stimulate microbial activity during secondary treatment’.

Line 49: What do the authors mean with "mixed"?

Thank you! We changed the phrase ‘mainly for primary and mixed’ to ‘mainly for primary and mixed primary and secondary sludges’.

Lines 69-70: The authors could explore further on the main ways of input of Pb into agricultural soils.

We added in the text: ‘In addition to natural sources such as igneous and sedimentary rocks, long-term use of pesticides and fertilizers is believed to be the main source of Pb contamination in agricultural soils [18]. Irrigation of agricultural land with domestic and industrial wastewater also contributes to the accumulation of Pb in the soil [19]’. Lines 72-75.

Line 103: Remove "regardless of Pb content". This is not significant for plants grown under Pd-contaminated soils.

Thank you! Done.

Lines 125-126: This is not true. This is only visible for Pb-free treatment.

You are right! We changed the phrase to ‘However, sludges only increased the leaf number of the plants grown on Pb-free soil, with no effect on plants grown on Pb-rich soil’.

Lines 129-130: This is not true. Replace increased with decreased.

Thank you very much! Guilty. We replaced ‘increased’ with ‘decreased’.

Lines 148-149: Do not present results that are not statistically significant.

Yes, thanks.

Lines 149-150: If this is one of the few truly significant results, it should be further explored and discussed.

We have added data (leaf mass per area, leaf nutrient content as a function of mass and area, photosynthetic nutrient use efficiency) to the manuscript to more clearly explain and discuss this result. Thank you.

In table 2, add a row with the values of 1-Rl:Rd.

Thank you. Done.

Lines 165: Replace affected with increased.

Sorry, but two-way Anova and, accordingly, Table 2 reflects only the degree of impact, and not its direction. The direction is discussed below.

Line 168: Do not describe trends that are not supported by statistical significance.

  1. We deleted part of the sentenses.

Line 170: "varied between 5 and 39%, with an average value of 17%." Why is this interesting to present?

Yes, this is not only interesting, but important. The magnitude of light inhibition of plant respiration (1−Rl:Rd) is important for modeling the carbon balance not only at the plant level, but to a greater extent at the level of ecosystems and the planet. Carbon fluxes and carbon balance calculations and predictions under changing climate is a current priority. Predictive models currently being developed in most cases do not take into account differences in the respiratory rates of vegetation in the light and in darkness. This results in an increase in the estimated daily carbon loss of the ecosystem, which of course introduces errores into carbon balance forecasting models. As it turned out in the last decade, the 1−Rl:Rd values are not constant and can vary depending on the type of plant and its growing conditions. Typically, stress increases the degree of slight inhibition of breathing. To expand our knowledge about the magnitude and degree of variability of this parameter, we present numerical values of 1−Rl:Rd in full confidence that they are in demand along with others.

Line 171: "highest values of Rl:Rd" This values were not significantly higher than in plants grown on Pb-free soil.

OK, we deleted this part.

Line 172-173: Not statistically significant.

OK, we removed this part.

Line 281: Please provide the diameter or area of each pot.

Done.

Line 288: How were pots irrigated? From the bottom or at the surface?

The surface.

Table 4 does not have any title.

Thank you! Sorry. Corrected.

Line 304: Harvested

Thanks!

Line 359-361: Most elements were not quantified in the plant tissues, right?

 You are right! Thanks!

Line 366: Two-way ANOVA, not One-way.

The correction is done.

Line 368-369: This conclusion seems a bit exaggerated considering the results presented. 

References should be formatted so that all species names appear italicized.

Thank you. Done.

All changes to the text are highlighted in yellow.

Thanks again for your comments and questions helped to improve the readability and clarity of the manuscript.

Best regards,

Reviewer 3 Report

Comments and Suggestions for Authors

Comments on Recovery effect of secondary

My first question is about secondary sludge. Typically such sludge has a primary treatment (basically filtration) and then a biological treatment that removes biological oxygen demand (typically aeration)/ I gather some biological treatment was used (line 297) and that would have decreased the BOM and presumably the organic matter. So how was there an increase in organic matter? Was ther water loss?

My next question wa about the design –biological measured had 5 replicated and chemical measured 3 replicates., so what was the experimental unit? Was it a pot? How ere the pots selected for measurement if they were not all assayed? Independence is a necessary assumption in most statistical analyses (including anova and non-parametric analyses).

Fundamental to such a design is blocking – blocking needs to be carried through during the experiment and into the chemical analysis phase.

A table of p values was not very helpful – you cannot ell if the effects are positive r negative.

The graphical presentations need to be improved – I enclose an example recent publication I reviewed. The ideas used there can be applied to the current data. That plot was created in the R package. That paper has now been

I do not understand how the addition of lead sulphate could have increased available K – more comments are needed about that.

There are phases in most experiments – here there was a glasshouse (pot) phase and hen there was a plant selection phase ofr the analyses and then a laboratory phase. The different phases of the trial must be coordinated – perhaps the same blocking structure maintained. Was this chevied? See Chis Brien et al. for details about this.

Comments on the Quality of English Language

Quite good -a correction noted on manuscript.

Author Response

First, we would like to thank the Reviewer for attentive revision of our manuscript and principal question.

Let me indicate the modifications made in the manuscript in the light of Reviewer’s comments.

My first question is about secondary sludge. Typically such sludge has a primary treatment (basically filtration) and then a biological treatment that removes biological oxygen demand (typically aeration)/ I gather some biological treatment was used (line 297) and that would have decreased the BOM and presumably the organic matter. So how was there an increase in organic matter? Was ther water loss?

In order to reduce the volume, chemical oxygen demand and biochemical oxygen demand, the primary sludge may undergo further treatment. This commonly involves biological decomposition through aerobic activated sludge systems, aeration and mixing to oxidise, or a successive combination of these or other methods to generate a more processed waste material known as secondary sludge.

Secondary sludges are enriched with organic matter in the form of dead microbial biomass, so secondary sludge application may result in an increase soil OM.

My next question wa about the design –biological measured had 5 replicated and chemical measured 3 replicates., so what was the experimental unit? Was it a pot? How ere the pots selected for measurement if they were not all assayed? Independence is a necessary assumption in most statistical analyses (including anova and non-parametric analyses).

Thank you. The text now explains the method for selecting plants and soil for measurements. We added to the manuscript:

Line 330:  Each unit treatment included eight pots.

Line 351-352: The leaves were randomly selected from plants from different pots of each treatment.

Line 387-390: The soils from five pots of each treatment was collected and mixed, and then three subsamples were used for chemical analysis. All plants of each treatment were collected from all pots, plant organs were mixed, and divided into three repetitions.

Fundamental to such a design is blocking – blocking needs to be carried through during the experiment and into the chemical analysis phase.

Yes, you are right.

A table of p values was not very helpful – you cannot ell if the effects are positive r negative.

Thank you! We have made changes to Table 2.

The graphical presentations need to be improved – I enclose an example recent publication I reviewed. The ideas used there can be applied to the current data. That plot was created in the R package. That paper has now been

Thanks a lot! The design you proposed is very successful! All Figures are improved.

I do not understand how the addition of lead sulphate could have increased available K – more comments are needed about that.

Unfortunately, we are currently unable to explain this result. Therefore, we decided to monitor potassium content in soils contaminated with heavy metals to confirm or refute this result. We removed potassium data from Table 1.

There are phases in most experiments – here there was a glasshouse (pot) phase and hen there was a plant selection phase ofr the analyses and then a laboratory phase. The different phases of the trial must be coordinated – perhaps the same blocking structure maintained. Was this chevied? See Chis Brien et al. for details about this.

Thank you very much for the recommendation. We have read the article

 ‘Brien, C.J.; Sermarini, R.A.; Borges Demétrio, C. Exposing the confounding in experimental designs to understand and evaluate them, and formulating linear mixed models for analyzing the data from a designed experiment. Biometrical Journal 2023, 65, 2200284.’

 but to be honest, we need time to fully understand the thoughts expressed in it.

All changes to the text are highlighted in yellow.

Thanks again for your comments and questions helped to improve the readability and clarity of the manuscript.

Best regards,

Round 2

Reviewer 2 Report

Comments and Suggestions for Authors

Thank you for considering my comments and suggestions.

I believe the final version of the manuscript is now suitable for publication.